# Hands-Free Human–Machine Interfaces Using Piezoelectric Sensors and Accelerometers for Simulated Wheelchair Control in Older Adults and People with Physical Disabilities

**DOI:** 10.3390/s25103037

**Published:** 2025-05-12

**Authors:** Charoenporn Bouyam, Nannaphat Siribunyaphat, Dollaporn Anopas, May Thu, Yunyong Punsawad

**Affiliations:** 1School of Informatics, Walailak University, Nakhon Si Thammarat 80160, Thailand; charoenporn.bo@wu.ac.th (C.B.); nannaphat.sir@wu.ac.th (N.S.); 2Informatics Innovative Center of Excellence, Walailak University, Nakhon Si Thammarat 80160, Thailand; 3Biodesign Innovation Center, Department of Parasitology, Faculty of Medicine Siriraj Hospital, Mahidol University, Bangkok 10700, Thailand; dollaporn.ano@mahidol.ac.th; 4Faculty of Engineering, Cambodia University of Technology and Science, Phnom Penh 121003, Cambodia; may.thu@camtech.edu.kh

**Keywords:** human–machine interface, face–machine interface, tongue–machine interface, piezoelectric sensor, assistive mobility devices, simulated wheelchair

## Abstract

Human–machine interface (HMI) systems are increasingly utilized to develop assistive technologies for individuals with disabilities and older adults. This study proposes two HMI systems using piezoelectric sensors to detect facial muscle activations from eye and tongue movements, and accelerometers to monitor head movements. This system enables hands-free wheelchair control for those with physical disabilities and speech impairments. A prototype wearable sensing device was also designed and implemented. Four commands can be generated using each sensor to steer the wheelchair. We conducted tests in offline and real-time scenarios to assess efficiency and usability among older volunteers. The head–machine interface achieved greater efficiency than the face–machine interface. The simulated wheelchair control tests showed that the head–machine interface typically required twice the time of joystick control, whereas the face–machine interface took approximately four times longer. Participants noted that the head-mounted wearable device was flexible and comfortable. Both modalities can be used for wheelchair control, especially the head–machine interface for patients retaining head movement. In severe cases, the face–machine interface can be used. Moreover, hybrid control can be employed to satisfy specific requirements. Compared to current commercial devices, the proposed HMIs provide lower costs, easier fabrication, and greater adaptability for real-world applications. We will further verify and improve the proposed devices for controlling a powered wheelchair, ensuring practical usability for people with paralysis and speech impairments.

## 1. Introduction

The World Health Organization (WHO) reports an annual increase in the global population of individuals with disabilities due to factors such as congenital conditions, aging, illness, and accidents. A significant challenge faced by individuals with disabilities is limited physical mobility [1], which can restrict their ability to move independently or communicate with others. The growing number of older adults worldwide may dramatically increase the number of people with mobility disabilities and impairments. Consequently, the development of assistive and rehabilitation devices is essential for enhancing their quality of life [2,3,4]. However, most commonly available technologies rely on controls such as buttons or joysticks, which may be inaccessible to those with severe disabilities. This highlights the need for inclusive technological advancements that accommodate the diverse needs of individuals with disabilities, posing a compelling challenge for researchers in designing innovative solutions that support independent living and social integration.

Human–machine interface (HMI) technology has become a critical area of research for developing assistive and rehabilitation devices for individuals with disabilities or the elderly [5,6]. HMI systems leverage biomedical and physical signals to facilitate machine control [6,7,8,9], with applications that include enabling control over electric wheelchairs through a variety of biomedical signals [10,11], such as facial muscle movements [12,13], head movements [14,15,16], eye tracking [17,18,19], electrooculography (EOG) [20,21,22], electromyography (EMG) [23,24,25], and electroencephalography (EEG) [26,27,28]. These studies aimed to empower individuals with severe disabilities to manage wheelchair movement independently. However, the widespread adoption of these systems has been hindered by their high cost and limited adaptability, rendering them inaccessible to many individuals with disabilities. Recent HMI research has increasingly focused on the development of more affordable and accurate systems. Innovations include the use of low-cost sensors, such as electromagnetic induction sensors [29,30], force sensors [31,32], and flex sensors [33,34], which can detect subtle movements in the tongue and facial muscles to generate control commands for electric wheelchair navigation. Despite advancements noted in previous research, several limitations remain in the sensor manufacturing process. These limitations include fabrication complexity, device durability, sensitivity, and the effects of temperature variations on measurement accuracy. Additionally, challenges continue to arise when deploying these sensors in various real-world environments, where system performance often fails to meet practical application requirements. Addressing these limitations is essential for improving the efficiency and reliability of HMI systems. For example, in our previous study [35] we utilized six thin-film piezoelectric sensors to capture facial muscle signals for electric wheelchair control. However, that approach had practical limitations, particularly regarding the inconvenience of attaching multiple sensors directly to the face.

This study introduces a wearable device with an adjustable design tailored to individual users, thereby enhancing its usability and feasibility. We reduced the number of piezoelectric sensors from six to three, while maintaining effective signal detection. Additionally, we utilized an accelerometer sensor to detect head movements, providing an alternative input for generating commands to control a simulated electric wheelchair. To evaluate its potential use for people with disabilities, we conducted experiments to verify the effectiveness of the proposed HMI system in older adult participants. The remainder of this paper is structured as follows: In Section 2, we demonstrate the prototype, which includes the wearable sensing device, face–machine interface method, head–machine interface method, and command translations. Section 3 presents the experimental results and discusses the verification of the efficiency of offline and online testing. Section 4 details the online testing procedures. Finally, Section 5 concludes and outlines future work regarding practical applications for individuals with physical disabilities and speech impairments.

## 2. Materials and Methods

### 2.1. Proposed Hands-Free HMI System Using Piezoelectric Sensors and Accelerometers for Wheelchair Control

To develop practical HMIs for hands-free control, we used displacement sensors to detect mechanical displacement resulting from human activities and interface machines through head, face, and tongue movements. Figure 1 illustrates the proposed hands-free HMI system designed for wheelchair control. This system consists of three primary components. The first is signal acquisition (shown on a blue background), which includes two key procedures: detecting facial muscle movements using a piezoelectric sensor and capturing head movements with an accelerometer, followed by the conversion of analog signals into digital signals for processing. Second, the signal analysis and classification process (depicted on a yellow background) integrates an algorithm for signal segmentation, a feature extraction process to gather essential data, and the subsequent classification of the signals. Finally, command translation for wheelchair control (represented on a green background) converts the output obtained from the classification into actionable commands for wheelchair operation. This approach can also be used in various other applications.

### 2.2. Prototyping

#### 2.2.1. Head-Mounted Device

Wearable devices are widely used in medical and healthcare applications to collect data from the human body for diagnostic and treatment purposes. They also contribute to the development of HMI systems [36,37,38]. In this study, we provided a head-mounted wearable device that uses piezoelectric sensors and accelerometers to acquire physical data from the face and head. We designed a comfortable and flexible head-mounted prototype using Shapr3D, as shown in Figure 2a. The prototype consists of three main parts: (1) a headset with a flexible shoulder containing two piezoelectric sensors at the left and right ends, (2) a flexible and rotatable shoulder containing a piezoelectric sensor that can connect to the end of either side of the headset, and (3) an accelerometer with a circuit container. The prototype model was 3D printed using a polylactic acid (PLA) filament, as depicted in Figure 2b,c. This design enables the adjustment of the sensor interface without removing the wearable device, according to the individual shape and size of the head.

#### 2.2.2. Sensing Modules and Processor

The electronic components of the head-mounted device can be divided into two parts: displacement sensor modules and the microprocessor, as shown in Figure 3. Each component is described in detail below.

Displacement sensor modules

For the face–machine interface, we used a piezoelectric vibration tapping sensor made from piezoelectric ceramic, which converts mechanical energy into electrical energy. Three 20.5 mm piezoelectric vibration tapping sensors were used for eye winking and tongue pushing. As described in Section 2.2.1, two piezoelectric sensors, PZ1 and PZ2, were placed on the left and right temples near the eye, respectively. PZ3 was placed on the left or right cheek, as shown in Figure 2b,c. These modules can provide both analog and digital signals. Analog piezoelectric signals were used as the input data for the microprocessor. We used the MPU6050 sensor module for the head–machine interface. This compact motion-tracking device has a 3-axis gyroscope, a 3-axis accelerometer, and a digital motion processor. It can communicate with microcontrollers via an I2C bus interface. According to the module installation in the head-mounted device, the x- and z-axes were used to detect head movements.

Microprocessor

We used an Arduino Nano ESP32 board as the microprocessor to process the input data from the sensing modules. The board is manufactured by Arduino S.r.l., based in Turin (Torino), Italy. This board features a NORA-W106 module from u-blox, equipped with a 32-bit ESP32 microcontroller. The Nano ESP32 supports Wi-Fi and Bluetooth wireless communication. We used a prototype shield board with a charger circuit and battery for plugging the sensor module and the ESP32 board. The ESP32 was used to convert analog piezoelectric signals to digital signals and acquire data from the MPU6050 sensor module. We programmed the ESP32 to collect and transmit data using the UART protocol through Bluetooth 5.0 modules. The modules were configured at a baud rate of 115,200 for serial communication with a computer, allowing for data processing and the implementation of algorithms. A 3.7 V/150 mAh lithium-ion battery is used to power the ESP32 board and sensor modules, providing approximately 45 min of continuous operation under normal conditions. Data processing was performed using Python (version 3.13.0). We utilized the Bluetooth Low Energy platform Agnostic Klient (Bleak) libraries (version 0.22.3) (https://pypi.org/project/bleak/, accessed on 21 January 2025) to read data from the head-mounted devices. The data were processed using the proposed algorithms and command translations. The commands were used to control simulated electric wheelchairs. We employed the McGill Immersive Wheelchair (miWe) simulator [39] to design experimental tasks that validate the effectiveness of the proposed online HMI system.

### 2.3. Proposed Actions and Commands

This study proposes a human–machine interface system that utilizes displacement sensing from a head-mounted device to generate commands based on face and head movements to control the direction of a wheelchair, as illustrated in Table 1. Using the proposed command, we created four directional control commands: turning left and right, and moving forward and backward. For the face–machine interface, we used eye and tongue movements. Winks in the left or right eye generated left- or right-turn commands, respectively. Blinking with both eyes created a command to move backward. Tongue pushes against the left or right cheek created a command to move forward. To develop the head–machine interface, we created a system that generates commands through head movements. The interface recognizes four distinct head-tilt directions (left, right, forward, and backward), each corresponding to a command for steering the wheelchair in these four directions. An example of the corresponding posture and actions is illustrated in Figure 3. In the idle state, a stop command is defined as a command without any action.

### 2.4. Proposed Algorithm

#### 2.4.1. Data Preprocessing

Data preprocessing was performed before the analysis and classification stages. For piezoelectric signal processing, a high-pass filter with a 3 Hz cutoff was used to remove low-frequency noise and baseline shifts caused by motion artifacts. A low-pass filter with an 80 Hz cutoff eliminated high-frequency noise and electrical interference, while a 50 Hz notch filter was applied to remove power line interference. For accelerometer data preprocessing, a 0.1–20 Hz bandpass filter was employed to focus on relevant head movements, removing low-frequency drift from slow movements and high-frequency noise. This process ensures that signal classification accurately captures movements of the eyes, tongue, and head.

#### 2.4.2. Face–Machine Interface

This study utilized piezoelectric sensors to measure muscle signals and generate commands for real-time processing. We applied conventional feature extraction techniques used in electromyography (EMG) to develop a method for detecting eye and tongue movements. Our approach utilizes time-domain features based on a research survey that focused on EMG-based machine control [39]. We used the maximum peak ($P_{MAX}$) feature derived from piezoelectric signals, which can be calculated using Equation (1).(1)$$P_{MAX}=\underset{x_{i}\in N}{\mathrm{Max}}\left(x_{i}\right)$$
where N is the signal acquisition dataset of each piezoelectric sensor and $x_{i}$ is the value at element i in the dataset.

The proposed classification algorithm for the face–machine interface consists of two main processing steps:(1)Calibration and Parameter Setting

Before testing, users must conduct a quick calibration by following the eye and tongue actions specified in Table 1 to activate each sensor for baseline parameter collection. Each action should be performed five times to compute the mean of the maximum values of five $P_{P\left(i\right)}$ values in Equation (2), which are calculated as follows:(2)$$B_{P\left(n\right)}=mean\left(P_{P\left(n\right)}\left(1\right),\;P_{P\left(n\right)}\left(2\right),\;P_{P\left(n\right)}\left(3\right),\;{P_{P\left(n\right)}\left(4\right),\;P}_{P\left(n\right)}\left(5\right)\right)$$
where $B_{P\left(n\right)}$ denotes the baseline parameters of the piezoelectric signals obtained from the piezoelectric sensors, i is defined as the index number of the piezoelectric sensors (n = 1, 2, 3), and $P_{P\left(n\right)}$ represents the maximum value of each action calculated using Equation (1). The baseline parameters were obtained for the threshold parameters $T_{s\left(n\right)}$ and $T_{P\left(1,2\right)}$ in Equations (3) and (4), respectively, which are calculated as follows:(3)$$T_{P\left(n\right)}=B_{P\left(n\right)}-0.25*B_{P\left(n\right)}$$(4)$$T_{P\left(1,2\right)}=(B_{P\left(1\right)}+B_{P\left(2\right)})-0.25*{(B}_{P\left(1\right)}+B_{P\left(2\right)})$$

During these actions, the feature parameters ($D_{P\left(i\right)}$) are defined by Equation (5) and calculated as follows:(5)$$D_{P\left(n\right)}=max\left(R_{P\left(n\right)}\left[1\right],\;R_{P\left(n\right)}\left[2\right],\ldots ,R_{P\left(n\right)}\left[N\right]\right)$$
where $R_{P\left(i\right)}$ represents the signals acquired in real time at a sampling rate of 1000 Hz. We used 1000 samples (N) for window sizes of 1 s, which were utilized for processing real-time features every 2 s: $E_{P\left(n\right)}$ and $E_{P\left(1,2\right)}$ in Equations (6) and (7), calculated as follows:(6)$$E_{P\left(n\right)}=\left\{\begin{array}{c} \left(D_{P\left(n\right)}-T_{P\left(n\right)}\right)\;\;\;\;\;\;,\;\;D_{P\left(n\right)}-T_{P\left(n\right)}>0\\ \;\;\;\;\;\;\;0\;\;\;\;\;\;\;\;\;\;\;\;\;\;\;\;\;\;\;\;\;\;\;\;,\;\;D_{P\left(n\right)}-T_{P\left(n\right)}<0 \end{array}\right.$$(7)$$E_{P\left(1,2\right)}=\left\{\begin{array}{c} \left(D_{P\left(1,2\right)}-T_{P\left(1,2\right)}\right)\;,\;\;D_{P\left(1,2\right)}-T_{P\left(1,2\right)}>0\\ \;\;\;\;\;\;\;0\;\;\;\;\;\;\;\;\;\;\;\;\;\;\;\;\;\;\;\;\;\;,\;\;D_{P\left(1,2\right)}-T_{P\left(1,2\right)}<0 \end{array}\right.$$

After obtaining the feature parameters, the “argmax” function determines the activated sensors within the output parameter (Op), as stated in Equation (8), which was calculated as follows:(8)$$O_{P}=\arg\underset{i}{\max}\left(E_{P\left(1\right),}E_{P\left(2\right)},E_{P\left(3\right)},E_{P\left(1,2\right)}\right)$$

(2)Decision Making and Command Translation

We propose command translation for simulated wheelchair control using facial muscle actions and human cognition. We directed the simulated wheelchair’s movement by combining eye and tongue actions (Table 1) to activate the target sensors. Tongue movements were used to move the wheelchair forward and backward using sensor PZ3. Winking the left eye activated sensor PZ1 to turn the wheelchair to the right. The left eye winking also activated sensor PZ2 to turn it to the left. Hence, we applied a decision rule based on the $O_{P}$ values, utilizing the index numbers of sensors (i) for four command classifications along with the idle state, as follows:$$\begin{array}{cc} \mathrm{if}\;O_{P}=1, & \mathrm{Decision}\;\mathrm{command}\;\mathrm{is}\;“\mathrm{Turn}\;\mathrm{Left}”\\ \mathrm{if}\;O_{P}=2, & \mathrm{Decision}\;\mathrm{command}\;\mathrm{is}\;“\mathrm{Turn}\;\mathrm{Right}”\\ \mathrm{if}\;O_{P}=3, & \mathrm{Decision}\;\mathrm{command}\;\mathrm{is}\;“\mathrm{Forward}”\\ \mathrm{if}\;O_{P}=4, & \mathrm{Decision}\;\mathrm{command}\;\mathrm{is}\;“\mathrm{Backward}”\\ Otherwise, & \mathrm{Decision}\;\mathrm{command}\;\mathrm{is}\;“\mathrm{Idle}” \end{array}$$

Figure 4 illustrates the proposed classification decision process for the face–machine interface system. For real-time processing, the system extracts feature parameters from three piezoelectric sensors that detect user actions and generate directional control commands every two seconds for the simulated wheelchair through keyboard interaction.

#### 2.4.3. Head–Machine Interface

The MPU6050 module was installed in a head-mounted device positioned in the y-plane. Hence, the angle data of the x- and z-axes were used to identify the direction of head tilt. The proposed classification algorithm for head movement detection consists of two main processing steps:(1)Calibration and Parameter Setting

Before use, the system requires a user calibration session to record the baseline acceleration along the x- and z-axes during head movements, following Table 1 for collecting baseline acceleration parameters. Each action should be performed three times to establish the maximum and minimum acceleration values along the x- and z-axes, $X_{max}$, $X_{min}$, $Z_{max}$, and $Z_{min}$, which are calculated as follows:(9)$$X_{max}=max\left(A_{L}\left[1\right],\;A_{L}\left[1\right],\ldots ,A_{L}\left[N\right]\right)$$(10)$$X_{min}=min\left(A_{R}\left[1\right],\;A_{R}\left[2\right],\ldots ,A_{R}\left[N\right]\right)$$(11)$$Z_{max}=max\left(A_{F}\left[1\right],\;A_{F}\left[2\right],\ldots ,A_{F}\left[N\right]\right)$$(12)$$Z_{min}=min\left(A_{B}\left[1\right],\;A_{B}\left[2\right],\ldots ,A_{B}\left[N\right]\right)$$
where $A_{L}$, $A_{R}$, $A_{F}$, and $A_{B}$ are the signals acquired in real time. We used 200 samples (N) for window sizes of 1 s each.

The baseline parameters in Equations (13)–(16) for the threshold values $T_{x-}$, $T_{x+}$, $T_{z-}$, and $T_{z+}$ were then established using 80 percent of the maximum and minimum accelerations along the x- and z-axes in Equations (9)–(12) to detect movement, which can be calculated as follows:(13)$$T_{x-}=X_{min}+0.8\left(\frac{\left(X_{max}-{\;X}_{min}\right)}{2}\right)$$(14)$$T_{x+}=X_{max}-0.8\left(\frac{\left(X_{max}-{\;X}_{min}\right)}{2}\right)$$(15)$$T_{z-}=Z_{min}+0.8\left(\frac{\left(Z_{max\;}-{\;Z}_{min}\right)}{2}\right)$$(16)$$T_{z+}=Z_{max}-0.8\left(\frac{\left(Z_{max}-{\;Z}_{min}\right)}{2}\right)$$

(2)Decision Making and Command Translation

The real-time acceleration data along the x- and z-axes, $A_{x}$ and $A_{z}$, were used to generate commands every 2 s based on threshold parameters. The classification process of the head–machine interface system is illustrated in Figure 5. Using a simple decision rule, the system categorizes the commands into four movement types, including an idle state, as follows:$$\begin{array}{ccc} \mathrm{if} & A_{x}<T_{x-} & \mathrm{Decision}\;\mathrm{command}\;\mathrm{is}\;“\mathrm{Turn}\;\mathrm{Left}”\\ \mathrm{else}\;\mathrm{if} & A_{x}>T_{x+} & \mathrm{Decision}\;\mathrm{command}\;\mathrm{is}\;“\mathrm{Turn}\;\mathrm{Right}”\\ \mathrm{else}\;\mathrm{if} & A_{z}<T_{z-} & \mathrm{Decision}\;\mathrm{command}\;\mathrm{is}\;“\mathrm{Forward}”\\ \mathrm{else}\;\mathrm{if} & A_{z}>T_{z+} & \mathrm{Decision}\;\mathrm{command}\;\mathrm{is}\;“\mathrm{Backward}”\\ \mathrm{else} & & \mathrm{Decision}\;\mathrm{command}\;\mathrm{is}\;“\mathrm{Idle} \end{array}$$

## 3. Results

### 3.1. Participants

This study focused on adults aged 60 to 69 years, regardless of gender, who were able to live independently and did not have neurological disorders affecting their mobility, in order to develop and validate the proposed HMI system in a controlled setting with older adults. The aim was to evaluate its efficiency, usability, and safety as an initial step. Participants excluded from the study included individuals with acute or chronic conditions that might hinder participation, those with impairments affecting facial, hand, or arm movements, and individuals who declined to provide informed consent.

Twelve older adults, six males and six females, aged between 60 and 69 years, with a mean age of 64.92 ± 3.95, participated in the study, as shown in Table 2. Additionally, all participants met the inclusion criteria, as none demonstrated impairments in facial muscles, bilateral hand function, or bilateral arm movement. Additionally, none had a history of neurological disorders affecting movement, such as stroke, Parkinson’s disease, or Amyotrophic Lateral Sclerosis (ALS). One participant had a lower limb impairment, and another reported weakness in the left (non-dominant) arm.

The participants participated in experiments to validate the effectiveness of the proposed system in controlling a simulated electric wheelchair. The experiment included two components: (1) assessing the effectiveness of the proposed HMIs for command creation, and (2) observing the performance of real-time control for the simulated power wheelchair. The participants reviewed the documentation and signed a consent form. This document ensures that personal information remains confidential and anonymous. The Ethics Committee of Human Research at Walailak University approved the research procedure involving human subjects (protocol code: WU-EC-IN-1-280-66) on 27 November 2023. This aligns with the ethical principles of the Declaration of Helsinki, the Council for International Organizations of Medical Sciences (CIOMS), and the World Health Organization (WHO).

### 3.2. Experiment I: Verification of the Proposed Hands-Free HMI System

In the first experiment, we evaluated the accuracy of command issuance for wheelchair control. All participants participated in the experiments without having any prior HMI experience. They used the proposed face– and head–machine interface and conventional joystick modalities and methods to generate commands, and the accuracy rate was measured. After completing the installation and calibration, each participant participated in a 20 min training session before starting the experiments. Each participant completed two trials, with 12 commands per trial, for 24 commands for each HMI modality. The command sequence is outlined in Table 3. Participants rested for 2 min before proceeding to the subsequent trial. After that trial, they rested for 5 min before beginning the following modality to avoid muscle fatigue.

We also utilized the assessment metrics of precision, sensitivity, and accuracy to evaluate the performance of the proposed actions and algorithms of the face– and head–machine interface systems for wheelchair control. These metrics can be calculated using Equations (17)–(19) as follows:(17)$$Precision=\frac{TP}{\left(TP+FP\right)}$$(18)$$Sensitivity=\frac{TP}{\left(TP+FN\right)}$$(19)$$Accuracy=\frac{\left(TP+TN\right)}{\left(TP+TN+FP+FN\right)}$$

A true positive (TP) indicates the correct result with the expected output. A true negative (TN) represents an accurate result for an unanticipated undesired outcome. A false positive (FP) refers to an inaccurately predicted result. False negatives (FNs) signify actual findings that were not expected.

Table 4 lists the accuracy values of command generation by comparing all three HMI modalities for each participant. With joystick control, the accuracy ranged from 95.8% to 100.0%, with a mean of 99.0%. The accuracy achieved with the proposed head–machine interface modality ranged from 87.5% to 100.0%, with a mean accuracy of 82.6%. The accuracy range obtained with the face–machine interface modality was 70.8% to 91.7%, with a mean accuracy of 82.3%. The efficiency of the head–machine interface system was similar to that of the joystick control. Most participants, except for Participants 7, 8, and 9, achieved a 100% accuracy rate when using both the joystick and head–machine interface modalities. The proposed head–machine interface can achieve higher accuracy than the face–machine interface. Nine participants achieved over 80% accuracy with the face–machine interface, which was lower than that in our previous work [35] using six thin-film piezoelectric sensors directly placed on the face to detect eye and tongue movements, where the accuracy exceeded 90%. This lower accuracy may have resulted from the inability of some participants to separately perform left and right winking.

Furthermore, the classification matrix for the proposed hands-free HMI systems based on the precision, sensitivity, and accuracy of each control command, as shown in Table 5 and Table 6, was used to evaluate the effectiveness of each HMI system. Table 5 presents the results of the proposed face–machine interface. These results demonstrate that the system can identify eye and tongue movements for the four command creations with an accuracy of over 90%. However, the success rate, precision, and sensitivity of the commands for turning left and right were relatively low. For some participants, winking with either the left or right eye was difficult and could lead to errors in activating commands. The forward and backward commands exhibited high precision, sensitivity, and accuracy. Most participants generated signal features by pushing their tongues against their left or right cheeks and blinking both eyes. Table 6 shows the results for the head–machine interface. The system had high classification efficiency for command creation, achieving a success rate, precision, sensitivity, and accuracy of over 95% for each command. These results indicate the significant potential of implementing our hands-free HMIs for online wheelchair control.

### 3.3. Experiment II: Performance of the Proposed Hands-Free HMI System for Real-Time Simulated Wheelchair Control

The same group of participants used the proposed HMIs and joystick (hand control) modalities to steer the simulated wheelchair to the goal and recorded the time taken. Each participant performed one round per control modality using two test routes. They wore the HMI device and sat in front of a computer to test the real-time steering of the simulated wheelchair, as shown in Figure 6. The first route in Figure 7a involved a path without obstacles, as depicted in Figure 7b. In the second route in Figure 8a, participants steered the wheelchair through a supermarket that contained obstacles, as illustrated in Figure 8a. Both routes had five checkpoints, labeled A to E, along with a designated starting point. Checkpoint E is the completion point for both routes. Participants began at the starting point and navigated the simulated wheelchair through a sequence of checkpoints, beginning at checkpoint A and continuing through B, C, and D, until reaching the final checkpoint E. Routes 1 and 2 must be completed within 3 and 6 min, respectively. If the participants could not reach the goal in time, the farthest checkpoint they reached was recorded. Before the experiments, each participant practiced controlling the simulated electric wheelchair using the HMI system for 15 min. Participants rested for 5 min before starting the next round. The recorded times taken were used to evaluate the effectiveness of the proposed HMI modalities and to assess user performance, as presented in Table 6 and Table 7.

Table 7 and Table 8 show the results of the efficiency comparisons between the proposed command translation patterns and the joystick based on the time spent steering the simulated wheelchair to complete Test Routes 1 and 2.

In Table 7, the results for Test Route 1 reveal that the time taken using the joystick control ranged from 40 to 95 s across all participants, with an average time of 59.0 s. For successful participants, the time taken with the proposed face–machine interface control ranged from 99 to 354 s, and the average was 241.8 s. The time with the proposed head–machine interface control ranged from 62 to 262 s, with an average time of 117.1 s. For Test Route 2 (Table 8), the time taken with the joystick control for all participants ranged from 67 to 160 s, and the average time was 110.3 s. The time taken with the proposed face–machine interface control ranged from 217 to 673 s, and the average time was 439.5 s for successful participants. The time with the proposed head–machine interface control ranged from 67 to 262 s, with an average time of 219.8 s.

Furthermore, in Test Route 1, Participant 1 spent the least amount of time steering with the face–machine interface at 99 s, while the time with the head–machine interface was 62 s. Participant 1 also spent the least amount of time steering on Test Route 2 with the face–machine interface at 217 s, while the head–machine interface took 97 s. The shortest times spent steering with the joystick control were 40 s for Test Route 1 and 67 s for Test Route 2.

The difference between the average time taken by the face–machine interface and joystick control on Test Route 1 was 182.8 s, whereas on Route 2, it was 329.2 s. The proposed face–machine interface control took approximately four times as long as the joystick control. The difference between the average time taken by the head–machine interface and joystick control on Test Route 1 was 58.1 s, whereas that on Route 2 was 109.5 s. The proposed head–machine interface control took approximately twice as long as the joystick control. The difference between the average time taken by the head–machine interface and joystick control on Test Route 1 was 58.1 s, whereas that on Route 2 was 109.5 s. The head–machine interface control took approximately twice as long as the joystick control. The difference between the average time taken by the face–machine interface and head–machine interface controls on Test Route 1 was 124.7 s, whereas on Test Route 2, it was 219.7 s. The proposed face–machine interface control took approximately twice as long as the head–machine interface control.

### 3.4. Satisfaction with the Proposed Hands-Free HMIs

After completing the experiments, all participants were required to complete a satisfaction questionnaire for each HMI modality. The questionnaire included seven questions (Table 9). Each question was evaluated using a five-point Likert scale ranging from 1 to 5, where 1 indicated “strongly disagree”, 2 indicated “disagree”, 3 indicated “neutral”, 4 indicated “agree”, and 5 indicated “strongly agree”. The participants were asked to respond to all questions thoughtfully.

Table 10 shows the responses to the satisfaction questionnaire provided by each participant. The analysis focused on usability and flexibility factors for developing and implementing hands-free HMIs for patients.

Regarding the ease of wearing the head-mounted device for both HMIs, the participants reported a high average satisfaction score of 4.58, ranging from 4 to 5, with a median score of 5, indicating that the device was easy to wear. The device offered a comfortable fit, as evidenced by a mean score of 4.50, ranging from 4 to 5, with a median score of 4. For ease of creating commands, the head–machine interface achieved a higher average score of 4.58, ranging from 4 to 5, compared to the face–machine interface, which averaged 3.58, ranging from 2 to 4. Additionally, participants reported that both the head–machine and face–machine interfaces were highly stable during operation, with average satisfaction scores of 4.25 and 4.5, respectively, ranging from 2 to 4, with a median of 4.

Moreover, the participants reported low fatigue levels for both modalities, with a median score of 5, indicating that they felt slightly fatigued after completing the experiment. The head–machine interface resulted in lower fatigue than the face–machine interface. In terms of confidence, participants gave the head–machine interface an average score of 4.83, with a median of 5, which was significantly higher than the face–machine interface’s average of 4.00 and median of 3. In terms of confidence, participants indicated that the head–machine interface received an average score of 4.83, with a median of 5, significantly higher than the face–machine interface, which scored 4.00 with a median of 3. Overall, satisfaction with both HMIs was high, ranging from 2 to 4. The head–machine interface received an average score of 4.83 and a median score of 5, which was higher than those of the face–machine interface, which had an average score of 4.00 and a median score of 4.

## 4. Discussion

According to the results of Experiment I, the proposed HMI systems demonstrated high efficiency among most participants, especially for the head–machine interface, which achieved an efficiency comparable to that of the joystick modality, as illustrated in Table 4. From the results in Table 5 and Table 6, the head–machine interface with accelerometer sensing achieved a success rate ranging from 95.8% to 100%, which was higher than that of the face–machine interface using facial muscle piezoelectric sensing, which ranged from 79.2% to 87.5%. Considering the individual commands of the face–machine interface, the forward and backward commands, activated by pushing the tongue against the cheek and blinking both eyes, yielded the highest success rates of 86.1% and 87.5%, respectively; however, these were still lower than the rate of 95.8% for the head–machine interface. Most participants found it easier to push their tongue against their cheek and blink both eyes than to wink each eye separately, influencing the efficiency of the left and right commands. For the head–machine interface, all participants demonstrated higher efficiency for the left and right commands by tilting the head compared to the forward and backward commands achieved by tilting the head forward and backward, which may have been influenced by a familiar level of head tilt. We could employ the advantages of each HMI to design a control strategy for a practical HMI system that covers all levels of physical disabilities.

Based on the results of Experiment II, the study with older adults, all participants completed the real-time simulated wheelchair control task on both test routes using the head–machine interface. However, nine participants utilized the face–machine interface for each test route because of the efficiency observed in Experiment I. Additionally, the face–machine interface required approximately twice as long to complete test routes 1 and 2 compared to the head–machine interface and took approximately four times longer than the joystick control. We found that the head–machine interface also provided high efficiency in real-time control, which required approximately twice as long as joystick control. None of the participants had experience in wheelchair control or HMI. We observed that face–machine interface control requires training sessions for beginning users to encourage them to achieve high efficiency. Moreover, we can select and recommend each command from both proposed hands-free HMIs for hybrid systems tailored to specific disability cases.

For the satisfaction results in Table 10, although the participants suggested that the prototype head-mounted device was easy to wear and comfortable for long periods during the experiments, it requires adjusting the shoulder to fit the face, which affects the efficiency and stability of the face–machine interface. Moreover, the participants reported that the face–machine interface required more effort to produce commands than the head–machine interface, which increased fatigue and lowered confidence in using the actual wheelchair control. Based on the overall score, participants expressed high satisfaction with the proposed hands-free HMIs, particularly regarding the head–machine interface modality.

### 4.1. Study Limitations

The proposed hands-free HMIs can be used by patients who retain movement in the head, face, or tongue to control actual powered wheelchairs. However, some limitations are as follows:(1)The study sample size is small, primarily aiming for a preliminary evaluation of the proposed head–machine interface (HMI) system with older adults for feasibility, usability, and safety before expanding to a larger, more diverse cohort.(2)The experiment was executed in a simulated environment, which may not precisely reflect real-world conditions.

### 4.2. Recommendations

Moreover, the preliminary experimental results indicate that the developed system can effectively control the movement of a simulated wheelchair. The recommendations for using the proposed HMIs to control wheelchairs are as follows:(1)A prototype of the head-mounted device must be designed and built for practical applications. The shoulder section, which houses piezoelectric sensors, should be flexible and shaped to fit the face.(2)Regarding hardware, the Arduino Nano ESP32 with Bluetooth was selected for rapid prototyping. A real-world implementation will need higher-performance hardware and faster communication.(3)The proposed hands-free HMIs require visual or audio feedback for training sessions and for creating real-time commands.(4)Implementing machine-learning techniques could enhance the classification accuracy of both hands-free HMIs.(5)The hands-free HMIs require validation for controlling powered wheelchairs in real environments. Additionally, validation in patients with quadriplegia should be performed.

### 4.3. Future Work

(1)Increasing the number of older participants and including individuals with quadriplegia in future studies will help enhance the reliability of the results and the applicability of hands-free HMI systems.(2)The efficiency of command generation can be improved by implementing machine-learning techniques, which may enhance classification accuracy.(3)We will interface the prototype with a real wheelchair to test its usability, reliability, and adaptability in real-world environments. Additionally, a powered wheelchair will be modified for semi-automatic operation with an integrated obstacle avoidance system.(4)We plan to upgrade the hardware to support more efficient real-time operations by enabling faster and more stable communication protocols, such as Wi-Fi and radio frequency (RF), along with increasing battery capacity for extended use.

## 5. Conclusions

This study aimed to develop hands-free HMIs for wheelchair control using facial muscle activity and head movements. The two HMI modalities include the face–machine interface, which uses piezoelectric sensing based on eye and tongue movements, and the head–machine interface, which measures head movements through an accelerometer. We prototyped and developed the head-mounted wearable device, implementing the proposed algorithms to generate four commands in real-time simulated wheelchair control experiments. Older adults participated in experiments to validate the hands-free HMIs. The results indicated that the efficiency of the head–machine interface was greater than that of the face–machine interface and comparable to that of joystick control. The head–machine interface required approximately twice as much time compared to joystick control for the tested routes, whereas the face–machine interface required four times as much time compared to joystick control. Both hands-free HMI modalities, especially the head–machine interface, can be used for simulated wheelchair control. In future work, we will further verify and enhance the proposed HMIs and head-mounted devices to control powered wheelchairs for practical use by patients and older adults.

## Figures and Tables

**Figure 1 sensors-25-03037-f001:**
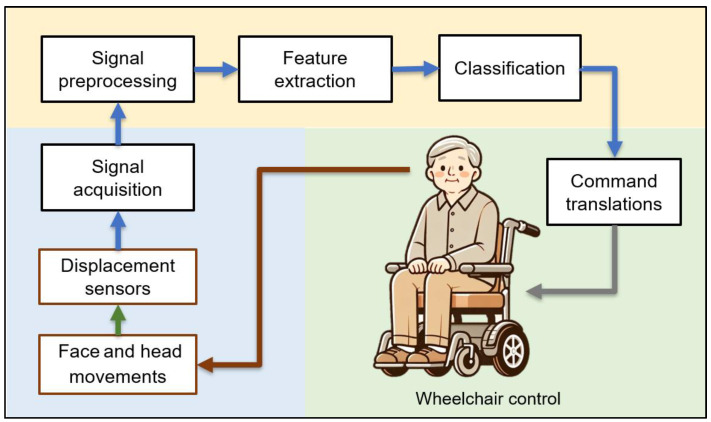
The overview of the proposed hands-free human–machine interface system for wheelchair control.

**Figure 2 sensors-25-03037-f002:**
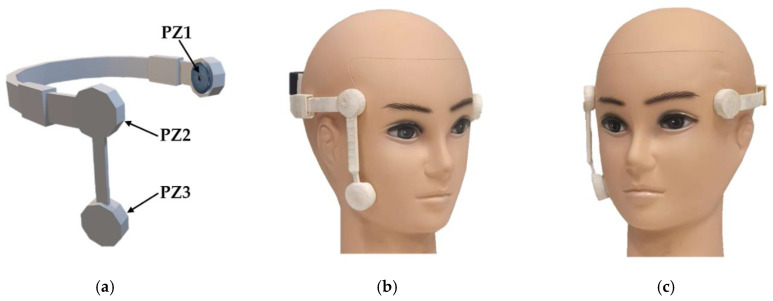
The prototype of the head-mounted device. (**a**) Three-dimensional modeling. (**b**,**c**) The 3D-printed head-mounted device prototypes worn on the head from the right and left perspectives.

**Figure 3 sensors-25-03037-f003:**
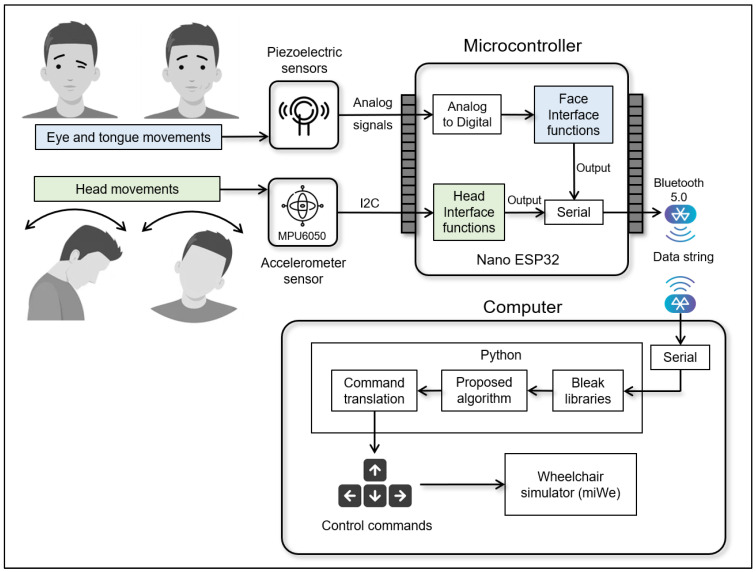
Diagram of the displacement sensing modules and connection for face– and head–machine interfaces.

**Figure 4 sensors-25-03037-f004:**
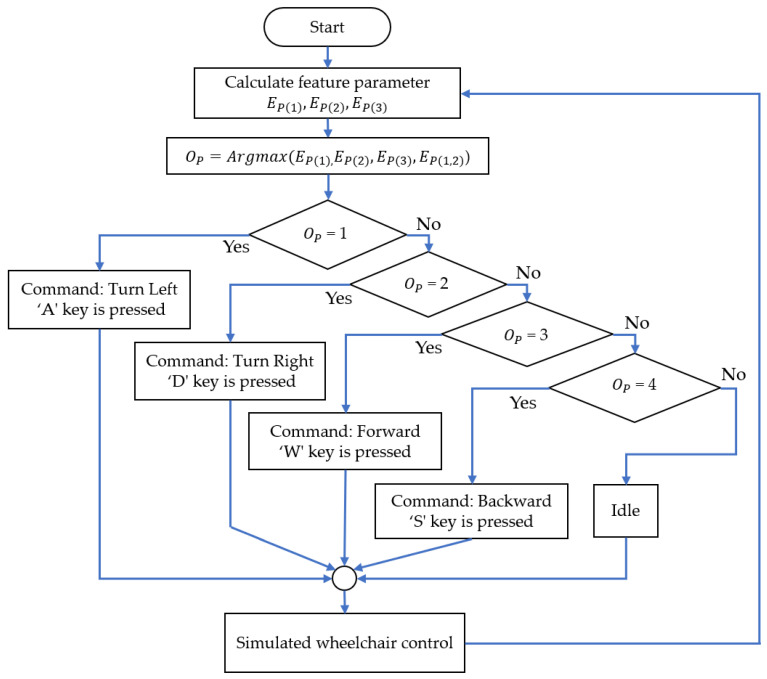
Flowchart of the classification process of the face–machine interface for simulated wheelchair control.

**Figure 5 sensors-25-03037-f005:**
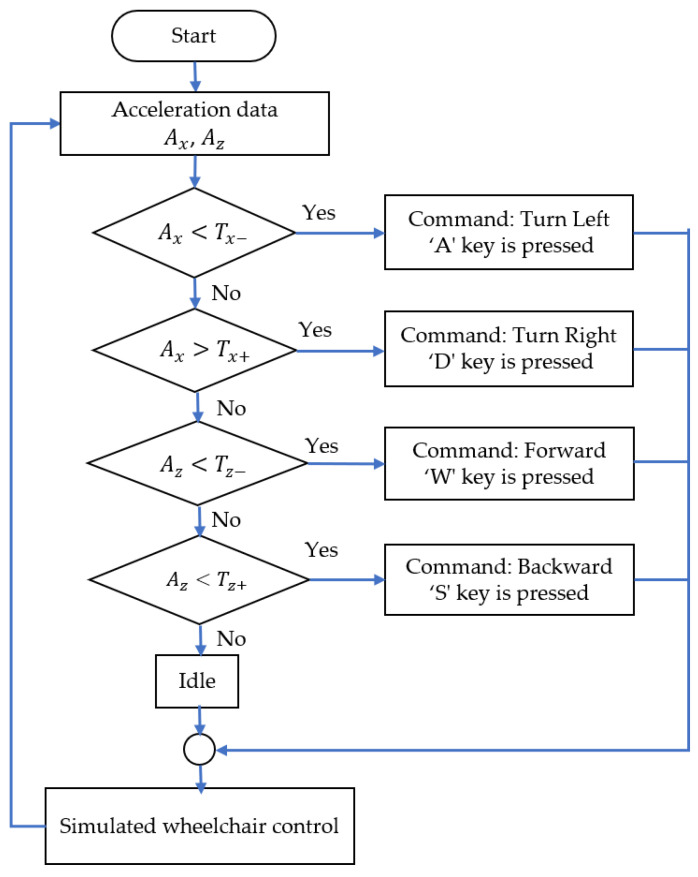
Flowchart of the classification process of the head–machine interface for simulated wheelchair control.

**Figure 6 sensors-25-03037-f006:**
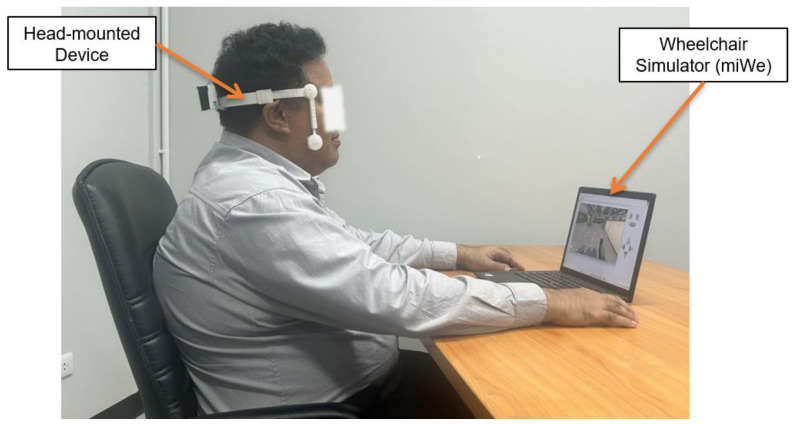
An example scenario of using the proposed HMIs to control a simulated wheelchair during the experiment.

**Figure 7 sensors-25-03037-f007:**
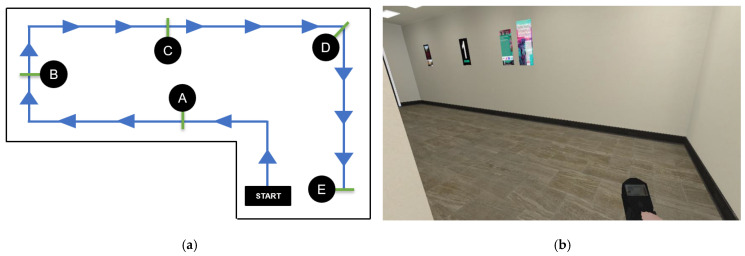
(**a**) Test Route 1 for controlling the simulated wheelchair in an obstacle-free environment, including five checkpoints (A to E) over a distance of 20 m. (**b**) A scenario encountered by the simulated electric wheelchair during testing in a corridor.

**Figure 8 sensors-25-03037-f008:**
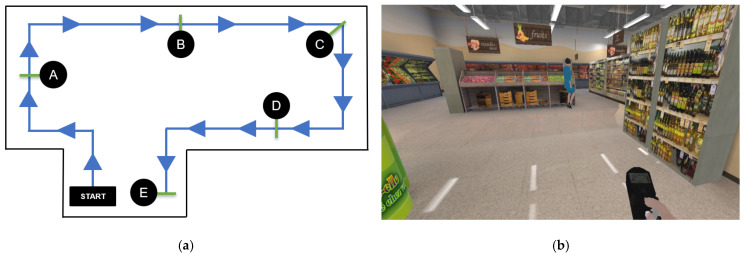
(**a**) Test Route 2 for controlling the simulated wheelchair in an environment with obstacles, including five checkpoints (A to E) over a distance of 35 m. (**b**) A scenario faced by the simulated electric wheelchair during testing in a supermarket.

**Table 1 sensors-25-03037-t001:** Proposed action mapping with commands for hands-free wheelchair control.

Command	Symbol	Face–Machine Interface		Head–Machine Interface	
		Actions	Target Sensors	Actions	Target Rotations
Turn Left	←	Winking the left eye.	PZ1	Tilting the head left.	(+) x-axis
Turn Right	→	Winking the right eye.	PZ2	Tilting the head right.	(−) x-axis
Forward	↑	Pushing the tongue on the left/right cheek.	PZ3	Tilting the head forward.	(+) z-axis
Backward	↓	Blinking both eyes.	PZ1 and PZ2	Tilting the head back.	(−) z-axis

**Table 2 sensors-25-03037-t002:** Participant demographics and physical condition.

Participant ID	Gender	Age (Years)	Physical Condition
1	Male	60	No impairment
2	Female	61	Lower limb impairment
3	Female	62	No impairment
4	Female	64	No impairment
5	Female	65	No impairment
6	Male	66	No impairment
7	Male	65	No impairment
8	Male	63	Left arm weakness
9	Female	69	No impairment
10	Male	69	No impairment
11	Male	67	No impairment
12	Female	68	No impairment

**Table 3 sensors-25-03037-t003:** Sequence of commands for evaluating the proposed HMI system.

Sequence No.	1	2	3	4	5	6	7	8	9	10	11	12
Commands	←	↑	→	↓	↑	→	←	↓	↑	←	↓	→

**Table 4 sensors-25-03037-t004:** The results of the proposed and conventional HMI modalities for all participants.

Participants	Average Classification Accuracy (%)		
	Joystick	Face–Machine Interface	Head–Machine Interface
1	100	91.7	100
2	100	83.3	100
3	100	75.0	100
4	100	87.5	100
5	100	70.8	100
6	100	87.5	100
7	95.8	83.3	91.7
8	95.8	87.5	91.7
9	95.8	83.3	87.5
10	100	87.5	100
11	100	70.8	100
12	100	91.7	100
Mean ± S.D.	99.0 ± 1.82	83.3± 7.32	97.6 ± 4.32

**Table 5 sensors-25-03037-t005:** Results of command classification of face–machine interface for wheelchair control.

	Output				Classification Metrics (%)			
Command	←	→	↑	↓	Success	Precision	Sensitivity	Accuracy
←	57	6	0	3	79.2	89.1	86.4	93.8
→	9	58	0	0	80.6	86.6	86.6	93.2
↑	0	0	62	0	86.1	100	100	100
↓	2	3	0	63	87.5	95.5	92.6	97.0
	Mean				83.3	98.0	97.9	99.0

**Table 6 sensors-25-03037-t006:** Results of command classification of head–machine interface for wheelchair control.

	Output				Classification Metrics (%)			
Command	←	→	↑	↓	Success	Precision	Sensitivity	Accuracy
←	69	0	0	2	98.6	100	98.6	99.7
→	1	69	0	2	100	94.7	100	98.6
↑	1	0	71	0	95.8	97.2	97.2	98.6
↓	0	0	0	72	95.8	100	95.8	99.0
	Mean				97.6	98.0	97.9	99.0

**Table 7 sensors-25-03037-t007:** The time taken and checkpoints completed by all participants in Route 1.

Participant	Joystick		Face–Machine Interface		Head–Machine Interface	
	Time (s)	Checkpoint	Time (s)	Checkpoint	Time (s)	Checkpoint
1	40	E	99	E	62	E
2	63	E	305	E	104	E
3	70	E	360 *	B	112	E
4	55	E	244	E	80	E
5	51	E	360 *	C	78	E
6	48	E	343	E	92	E
7	57	E	343	E	191	E
8	62	E	143	E	117	E
9	53	E	354	E	262	E
10	49	E	202	E	78	E
11	95	E	360 *	A	147	E
12	65	E	143	E	82	E
Mean ± S.D.	59.0 ± 14.1		241.8 ± 77.8 ^†^		117.1 ± 57.9	

Note that * indicates that the participant could not complete the task within the specified time. ^†^ is the average time without *.

**Table 8 sensors-25-03037-t008:** The time taken and checkpoints completed by all participants in Route 2.

Participant	Joystick		Face–Machine Interface		Head–Machine Interface	
	Time (s)	Checkpoint	Time (s)	Checkpoint	Time (s)	Checkpoint
1	67	E	217	E	97	E
2	118	E	720 *	D	198	E
3	160	E	720 *	C	242	E
4	105	E	405	E	154	E
5	107	E	720 *	B	190	E
6	110	E	380	E	165	E
7	103	E	380	E	353	E
8	134	E	494	E	146	E
9	94	E	673	E	532	E
10	88	E	536	E	193	E
11	136	E	720 *	B	193	E
12	102	E	431	E	174	E
Mean ± S.D.	110.3 ± 24.4		439.5 ± 133.4 ^†^		219.8 ± 116.2	

Note that * indicates that the participant could not complete the task within the specified time. ^†^ is the average time without *.

**Table 9 sensors-25-03037-t009:** The satisfaction questionnaire.

Questions No.	Survey Statements
1	I can easily wear the device.
2	I feel comfortable while wearing the device.
3	I can easily create commands for steering the simulated wheelchair.
4	The HMI device and system demonstrate stability during operation.
5	I feel slightly fatigued after completing the experiment.
6	I feel confident operating the electric wheelchair with the device.
7	Overall, I am satisfied with my HMI system experience.

**Table 10 sensors-25-03037-t010:** Results of satisfaction with the proposed HMIs for wheelchair control.

Question No	HMI Modalities	Satisfaction Score		
		Range	Median	Mean ± S.D.
1	Face–machine interface	4–5	5	4.58 ± 0.51
	Head–machine interface			
2	Face–machine interface	4–5	4	4.50 ± 0.52
	Head–machine interface			
3	Face–machine interface	2–4	4	3.58 ± 0.79
	Head–machine interface	3–5	5	4.42 ± 0.90
4	Face–machine interface	3–5	4	4.25 ± 0.75
	Head–machine interface	3–5	4	4.50 ± 0.43
6	Face–machine interface	3–5	5	4.42 ± 0.90
	Head–machine interface	4–5	5	4.92 ± 0.29
5	Face–machine interface	3–5	3	3.67 ± 0.89
	Head–machine interface	3–5	5	4.42 ± 0.79
7	Face–machine interface	3–5	4	4.00 ± 0.90
	Head–machine interface	3–5	5	4.83 ± 0.58

## Data Availability

The data presented in this study are available upon request.

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
