# Peer review of "Hands-Free Human–Machine Interfaces Using Piezoelectric Sensors and Accelerometers for Simulated Wheelchair Control in Older Adults and People with Physical Disabilities"

_sensors, 2025, doi:10.3390/s25103037_

Round 1

Reviewer 1 Report

Comments and Suggestions for Authors

This is a well-structured and relevant paper that presents a hands-free human-machine interface device using piezoelectric sensors and accelerometers to control a wheelchair. The proposed system aims to assist elderly individuals and people with physical disabilities, and the authors have demonstrated solid progress in both the design and implementation phases. The prototype is clearly described, and its components are well explained. Furthermore, the authors provide evidence of iterative improvements, showing continuity with previous work.

  • The calibration procedures and command translation mechanisms are clearly detailed for both the face-machine and head-machine interfaces.
  • The testing procedures are well described, and the use of performance metrics such as precision, sensitivity, and accuracy is appropriate to validate the system's effectiveness.
  • The inclusion of user satisfaction questionnaires adds significant value by incorporating subjective assessment and user-centered design principles.
  • The motivation behind the project is well justified.

Even though the article has many strenghs, it could benefit from some imrpovements. Below I listed my suggestions and comments:

  • The abstract should better highlight how the proposed prototype addresses gaps in existing solutions and how it differs from commercial devices.
  • Clarify inconsistencies in the terminology used. For example, terms such as “piezoelectric” and “displacement sensors” are used interchangeably, which may cause confusion. Are they referring to the same sensor technology?
  • The abstract mentions the use of the tongue, but later the paper refers to eye winking and facial movement. This inconsistency in sensor input modalities should be clarified.
  • The yellow box description in Figure 1 refers to piezoelectric sensors and facial movements but does not clearly address how head movement or displacement is captured. Further clarification is needed on the role of each sensor type.
  • If accelerometers and other sensors used also qualify as displacement sensors, it is worth questioning why the title only mentions piezoelectric sensors.
  • The paper would benefit from a discussion about the energy requirements and battery life of the prototype. Autonomy is an essential factor for real-world usability.
  • The sample size of 12 participants is relatively small. The authors should justify this limitation and ideally include a dedicated section on Study Limitations.
  • Inclusion and exclusion criteria for participants should be clearly stated.
  • It would also be helpful to include a table of participant demographics, with details such as age, gender, physical condition, and how these demographics reflect the target population (older adults and individuals with physical disabilities).
  • The age range mentioned (60–69) may not fully represent the broader older adult population. Were participants with physical disabilities included in the sample?
  • The Discussion should be a standalone section.
  • The manuscript would be improved by adding sections for Study Limitations, Recommendations, and Future Work.
  • A suggestion for future development: Have the authors considered constructing a complete wheelchair prototype and testing it in real-world environments?

Reviewer 2 Report

Comments and Suggestions for Authors

It is of utmost importance to consider that the title adds that the control of the wheelchair was reached up to the simulation version and no experimentation was carried out with a wheelchair built for this purpose.

An explanation of how sensor data is processed and whether data preprocessing is necessary to achieve optimal results is required.

The methodology needs to be improved by adding flowcharts explaining the algorithm programming for the Face-Machine Interface and Head-Machine Interface. The classification process is based on interaction with the simulation platform.

It should be clearly mentioned how the experiments shown in tables 6 and 7 were performed, for each checkpoint for Joystick Face-machine interface and Head-machine interface.

Among the conclusions, it should be noted that only the control stage of a simulated system was reached.

Round 2

Reviewer 1 Report

Comments and Suggestions for Authors

I have completed the second review of the revised manuscript. The authors have thoroughly addressed all the previous comments and made the necessary improvements. I now consider the manuscript suitable for publication in its current form.